# Methods to Reduce Energy and Polymer Consumption for Fused Filament Fabrication 3D Printing

**DOI:** 10.3390/polym15081874

**Published:** 2023-04-13

**Authors:** Owen James Harding, Christian Andrew Griffiths, Andrew Rees, Dimitrios Pletsas

**Affiliations:** College of Engineering, Swansea University, Swansea SA2 8PP, UK

**Keywords:** 3D printing, fused filament fabrication, fused deposition modelling, power efficiency, manufacturing optimisation, polylactic acid (PLA)

## Abstract

Fused Filament Fabrication (FFF) 3D printing is an additive technology used to manufacture parts. Used in the engineering industry for prototyping polymetric parts, this disruptive technology has been adopted commercially and there are affordable printers on the market that allow for at-home printing. This paper examines six methods of reducing the energy and material consumption of 3D printing. Using different commercial printers, each approach was investigated experimentally, and the potential savings were quantified. The modification most effective at reducing energy consumption was the hot-end insulation, with savings of 33.8–30.63%, followed by the sealed enclosure, yielding an average power reduction of 18%. For material, the most influential change was noted using ‘lightning infill’, reducing material consumption by 51%. The methodology includes a combined energy- and material-saving approach in the production of a referenceable ‘Utah Teapot’ sample object. Using combined techniques on the Utah Teapot print, the material consumption was reduced by values between 55.8% and 56.4%, and power consumption was reduced by 29% to 38%. The implementation of a data-logging system allowed us to identify significant thermal management and material usage opportunities to minimise power consumption, providing solutions for a more positive impact on the sustainable manufacturing of 3D printed parts.

## 1. Introduction

The UK’s net-zero [1] target for greenhouse emissions by 2050, enshrined in the European Green Deal, [2] recognises the climate crisis. Consequently, manufacturing processes are being continuously re-examined to both improve their economic performance and minimise their environmental impact. Environmental improvement often takes a two-pronged approach: (i) to reduce the amount of materials that are consumed (or wasted) as part of the industrial process and (ii) to reduce the amount of energy required in the manufacturing process. One route to minimising energy use is through optimised product manufacturing [3,4]. Fused Filament Fabrication (FFF), a form of additive manufacturing (AM), shows significant room for improvement in terms of economic viability and environmental impact [5]. This paper investigates the potential energy and material savings when building polymer components using this technology.

AM is gaining vital growth because it is a technique capable of producing complex structures with ease. More industries are adopting AM technologies to accelerate product development alongside improved cost-effectiveness [6]. There was a widening scope for AM technologies during the COVID-19 pandemic, and a dramatic increase in the demand for medical devices such as ventilators and personal protective equipment, and 3D printing can help produce many of these products [7]. During the pandemic, supply chains for medical health care devices, personal protective equipment, raw materials, and food processing and distribution were disrupted, caused by an increased demand for medical and health equipment [8]. The additive manufacturing community took initiatives to provide a stopgap regarding the lack of medical devices and kits. To ensure patients’ health, various firms and institutes emerged with radical design changes and adaptive technologies, employing additive manufacturing for end products and holding clearance from regulators and industry standards [9]. During this time, the market share price and investor interest surged by 70%, exhibiting an increased demand for the 3D printing market [8]. FFF, also known as the material extrusion AM technique, utilises polymers extruded into layers on top of one another to form an object. It was first introduced in 1992 by American company, Stratasys [10,11]. The method was originally used to create conceptual models to aid product design. Since then, FFF has matured enough to produce test parts optimised for the production of end-use parts [12,13]. In terms of sustainable manufacturing, the FFF printing parameters, namely, infill density, layer thickness, feed rate, shell thickness and build plate adhesion, influence the energy usage. The specific energy and specific time both decrease by decreasing the infill density and shell thickness and by increasing the build feed rate and layer thickness [3]. The more complex a product is, the more advantageous 3D printing it will be since the cost and energy for printing is not dependent on component complexity [14].

AM has been recognised as an energy-intensive technique due to the amount of energy required to form the multiple thin layers. The energy consumption per unit volume of material is generally much higher than other manufacturing techniques, but when evaluating environmental performance of AM it has been observed that it has the potential to be more environmentally friendly than conventional manufacturing methods [15,16,17]. Some work has been performed to establish the energy impact of machine architecture and process effects on power consumption [18,19], but for FFF, limited efforts have been made in this direction in energy consumption. A method of heating the polymer to the desired temperature (*T*) and controlling the flow rate and an accurate mechanical method of aligning and moving the extrusion point are required [20]. The general fabrication process of FFF can be divided into the following stages: Extruder heating, Bed heating, Levelling, Printing and Bed cooling [21]. It has been observed that power demand is not constant and changes across during the fabrication process, and moreover, the power demand pattern can be adjusted by changing process parameters [21]. Since most of the power consumed by a 3D printer is via the heated bed and extruder (72.9%), minimising thermal loss from these areas is our main area of focus [22]. For the first two stages, the extruder and bed heating insulation have the most potential to reduce heat loss.

High energy consumption and a slow printing process are a disadvantage for 3D printing [23]. Additionally, for FFF there are many ways for a print to fail [24], and 34% of the plastic used is wasted with failed prints. This waste is both an environmental and financial concern, with just 39% of plastic waste being recycled locally in the UK and the rest generally being exported to other nations [2]. Optimisation strategies can improve the reliability of AM. However, the environmental impact of this production method is still unclear due to many influencing factors [25,26]. Insufficient research is conducted on energy use, and the impact of accumulative energy has been identified as important if future studies are not conducted [27].

The objectives of this paper are to reduce unnecessary polymer usage and reduce energy consumed when building FFF parts. The two main methods of reducing the energy consumption are heat insulation and print volume. Specifically, heated bed insulation, hot-end nozzle insulation and a full 3D printer enclosure are used to assess the influence of insulation. This is followed by modifying the volume of the test part build using reduced support line width and optimised infill, using the new lightning internal support builder and a foaming polymer. Each of these is assessed, and the potential savings are quantified. The paper layout is as follows: the following section is the experimental section, which is followed by the results and a discussion of the tests. Finally, conclusions are drawn about the energy savings observed from the performed experiments.

## 2. Experimental

### 2.1. Experiments

For all three machines, the following ten experiments were performed:Energy measurement:
Heated bed influence on energy results. All three machines will build parts with and without bed insulation and the energy will be measured for each.Hot-end influence on energy. All three machines will build parts with and without nozzle insulation and the energy will be measured for each.Printer enclosure influence on energy. All three machines will build parts with and without the enclosure and the energy will be measured for each.For all three machines, an extended print test (>8 h) will be performed to observe the temperature differential inside and outside the printer enclosure.Support line width influence on energy. All three machines will build parts with two different line widths and the energy will be measured for each.Infill influence on energy. All three machines will build parts with two different infill types and the energy will be measured for each.Optimised energy results. All three machines will build parts with optimised insulation, line width and foaming PLA.
Build time and Material measurement:
8.Support line width influence on time and material results. All three machines will build parts with two different line widths and the time and material will be measured for each.9.Infill influence on time and material results. All three machines will build parts with two different infill types and the time and material will be measured for each.10.Optimised material and time results. All three machines will build parts with optimised insulation, line width and foaming PLA.

### 2.2. Pre-Trials

Pre-trials confirmed that the experimental approach of repeated responses for part weight and energy consumption were consistent. However, to validate this, a design of experiments approach was performed. Screening experiments using an ISO 527-2-1A dog bone test part were performed; the control factors included PLA and LW-PLA materials and the process factors were layer height, infill, print speed and extrusion temperature. To establish that the process was in control, the SE Mean (standard error S^2^), Variance (σ^2^) and the Coefficient of Variation (COV) were investigated (Table 1). The results were consistent and the reproducibility shows that the process was optimised for the ten experiments.

### 2.3. 3D Printers

The 3D printers used for testing were of the ‘Ender’ series from Shenzhen Creality 3D Technology Co, Ltd (CN). The three selected were the Ender-2 Pro, Ender-3 Pro, and Ender-5 Pro. These were selected for their range of gantry configurations, with a bed surface area of 484 cm^2^ for the Ender-3 and Ender-5, and 272.25 cm^2^ for the Ender-2. To ensure consistency, the layer slicer used was the Ultimaker CURA version 4.12 (NL), with default settings used for each printer. All tests used 0.6 mm diameter brass nozzles for consistency. The control filament used was polylactic acid (PLA), supplied by colorFabb (NL). The filament used to investigate the potential polymer savings provided by the foaming agents within the filament was colorFabb Natural LW-PLA. The ambient conditions were maintained by a thermostatically controlled electrical heater set to 25 °C.

### 2.4. Power Consumption Tests

Most of the power consumed by a 3D printer is via the heated bed and extruder [22] and the greatest opportunity to reduce power consumption resides with the insulation of the thermal components. To record the power consumption and establish the thermal loss from these areas, PZEM-017 dataloggers (CN) were used. They provided power consumption measurements per second for whichever components they were connected to, with ±1% inaccuracy [28]. Each was wired as recommended by cutting the heater cartridge cables of the component that readings were to be taken from. These leads were connected to the PZEM-017 and a 50A shunt to create a closed circuit. The PZEM-017 was powered from a separate micro-USB cable not connected to the printer being tested, with data being streamed via the RS485 port. An RS485 to USB-A converter was used to allow for the readings to be directly recorded.

### 2.5. Heated Bed Insulation

The heated bed of a 3D printer is the surface upon which the initial layers of a print are laid. The underside of each printer had approximately 15 mm of clearance. Since this surface was in direct contact with air, it was likely that savings by reducing thermal loss from this area would be significant. The insulation selected was 12 mm thick CC400 VAF-S self-adhesive barafoam. It is rated to withstand temperatures of 100 °C, equivalent to the maximum heated bed operating temperature. Additionally, it has a low thermal conductivity of 0.053 W/mK, demonstrating its capability to reduce thermal transfer [29]. To test the thermal loss before any insulation had been applied, a PZEM-017 was fitted to the heated bed of each 3D printer. Having turned on the electric heater and established that the ambient temperature had reached 25 °C, the printers were turned on and the bed temperature of each was manually set to 60 °C. Once all of them had achieved and sustained a bed temperature of 60 °C for at least 60 s, a timer of 8 h was started as the dataloggers were set to start recording results. A testing time of 8 h was selected to minimise the effects of spikes or troughs in the power consumption. These fluctuations are caused by the negative feedback loop that the motherboard uses to switch the heater from on to off based on the thermistor temperature. Therefore, an average consumption over a longer period of time is a more representative indicator of energy loss throughout a print. Upon completion of the 8 h period, the printers were turned off, and the recorded information was used to discover the average bed power consumption. To assess the effectiveness of this insulation, the methodology described earlier was repeated with the 12 mm thick barafoam in place.

### 2.6. Hot-End Insulation

The extruder hot-end is where filament is heated to a desired temperature to ensure sufficient extrusion through the nozzle. The selected printers in this research used an insulated silicone ‘sock’ that covered the hot-end. This helped to mitigate the dissipation of heat from the hot-end ‘block’ to the surrounding air. The one provided by Creality consisted of 1.5 mm thick silicone, which covered two-thirds of the heater block surfaces and the base of the nozzle. Before insulation testing was conducted, each printer was individually turned on and the heat block was set to reach 230 °C (a hot-end temperature of 230 °C was chosen to allow for these results to be incorporated and analysed alongside the foaming polymer testing. As the foaming PLA foams between 215 °C and 250 °C, 230 °C allows for the foaming filament to have potential impact in its testing while allowing results to be analogous to normal 3D printing settings). Testing was conducted by setting up a PZEM-017 datalogger and an 8 h timer was set to start recording power consumption readings. After the time elapsed, the data were saved before turning the printers off, applying the Creality silicone ‘sock’ insulator and repeating the process.

### 2.7. Full 3D Printer Enclosure

A 3D printer enclosure is characterised by a container within which the 3D printer is to be sealed inside of during operation. The concept was initially patented by Stratasys Inc., although these patents expired in February 2021 [30]. Several companies have created their own commercially available enclosures. Enclosures primarily offer features such as being able to mitigate the spread of fumes, helping to control potential fires and reduce the noise of a 3D printer operating. However, enclosures also reduce wasted thermal energy by trapping the air heated by the printer within a smaller volume. This in turn raises the temperature of the internal volume above that of the ambient air outside the enclosure, resulting in a reduced temperature gradient between the 3D printer components and the surroundings. Before evaluating the enclosure, the nozzle replacement and the hot-ends were cleaned using the process described previously in Section 2.4. Two PZEM-017s were used for each printer; one was attached to the heated bed and the other to the heater cartridge within the hot-end. Once the hot-end reached 230 °C, the heated bed 60 °C and the ambient temperature 25 °C, an 8 h timer was started as the dataloggers recorded power consumption readings. Once control testing had been completed, each 3D printer was put into a Creality enclosure, and a thermometer probe was set in a fixed position at the centre of each enclosure. The thermometer selected for this had a digital LCD display, which remained outside the enclosure to allow for easier recording. The same control testing methodology was repeated, although the internal enclosure temperature was regularly recorded. The internal enclosure temperature was noted every 15 min after the start of the 8 h timer. The readings from the datalogger attached to the heated bed and those from the heater cartridge in the hot-end were combined to find the average power over the entire time period.

### 2.8. Test Part and Printer Settings

To form combined testing, a testing model known as the ‘Utah Teapot’ was selected. It was chosen due to its extensive historical usage within 3D modelling and animation as a reference object. The model is automatically imported the right way up with the flat base of the teapot resting evenly on the centre of the build plate. The control test was conducted without any modifications applied to any of the 3D printers. A G-code for the teapot, using the settings in Table 2, a 20% cubic infill, and a support line width of 0.6 mm, was created for each printer (Figure 1). A PZEM-017 was fitted to each 3D printer to record total power consumption. This was run on each unmodified printer using colorFabb black economy PLA with an ambient temperature of 25 °C.

### 2.9. Polymer Consumption Tests

The specific testing methodologies of the material were independent from one another and did not include any of the power consumption modifications. The control set of slicing parameters for the printers were the Ultimaker CURA recommended default settings. For each experiment, a new 0.6 mm brass nozzle was used, to remove nozzle wear as a potential factor impacting material usage. It needed to be established if the consumption modifications had an effect on the 3D printer power consumption, so 3D printer system power consumption was recorded by wiring a PZEM-017 between the 3D printer power supply and control motherboard. Additionally, a plug-in digital LCD power reader was inserted between the 3D printer kettle lead and the building power outlet. These provided live wattage readings in addition to recording the cumulative energy consumed since the meter had been last reset.

### 2.10. Reduced Support Line Width

In FFF 3D printing, ‘support’ refers to the creation of sacrificial material to provide a surface upon which further layers can be printed. Since support will be discarded, minimising the amount of polymer required reduces waste. An Ultimaker CURA was used to create the object g-code for each 3D printer. A test cube object with equivalent side lengths of 50 mm was imported into CURA and was set to ‘generate support’ for that volume. For each printer, two g-code files were generated: one with a support line width of 0.6 mm, the other with a support line width of 0.4 mm. The default support line distance of 3.0 mm and support pattern of ‘zig-zag’ remained unchanged. Tests were conducted by removing the magnetic build plate from each 3D printer, removing any residue from prior prints, and then weighing and recording its mass with a digital scale. The build plates were replaced back on each respective heated bed, and the thermostatically controlled heater was turned on. The g-code for a support line width of 0.6 mm using the default settings (Table 1) was started alongside the datalogger readings when the ambient temperature reached a stable 25 °C. Upon completion of the print, the build plate was re-weighed, being careful not to lose any of the polymer deposited upon it. The difference between the two mass readings reflects the total mass of filament consumed. This process was then repeated using a g-code where the support line width was reduced to 0.4 mm.

### 2.11. Lightning Infill

The default 3D printing infill pattern within the Ultimaker CURA was ‘cubic’, where a uniform internal cuboid structure was formed throughout the model. This infill provides a surface upon which upper ‘top’ layers may be placed. However, this infill is also placed in areas where a successful print could be completed without it [31]. A lightning infill pattern is different to a cubic one; it only forms internal structures in areas required to prevent print failure, leaving areas that are not in need of hollow support [32] (Figure 2). It is named lightning infill due to its branching structure, which resembles a lightning bolt. This is ideal for cosmetic models that do not require significant structural integrity. It may also be applied to models that have mechanical application, allowing for the polymer that would have been used on the internal infill to be shifted to areas where its use is most critical. A front-facing and top-down X-ray view of both internal structures show that when using lightning infill, the material usage is reduced while maintaining the same amount of contact for upper surfaces. Testing was completed via the same procedure described previously for the reduced support line width, although different g-codes were used instead. A g-code was created for both infill types at 20% density using the other default settings in Table 1. Support generation was turned off since the 50 mm cube would not require any external supports. Upon completion, the printed samples were sawn in half to verify the infill pattern. 

### 2.12. Foaming Polymer

Foaming filaments are 3D printing polymers that have foaming agents dispersed within them. While the hot-end temperature is above the activation temperature, the foaming agent generates gas, dispersing bubbles within the deposited material. The amount of foaming varies with printing temperature between 215 °C and 250 °C [33]. This allows for a reduction in part density and subsequently the amount of filament consumed by the printer. The datasheet of foaming filament Natural LW-PLA, supplied by colorFabb, says that at the maximum recommended printing temperature of 250 °C the foaming is sufficient to cause the printed filament volume to increase by 185%. This allows for a reduced polymer flow of 35% while still forming a successful print. However, this may impact the structural strength of the final part. Lower amounts of foaming (amounts less than 15%) have been shown to have minimal impact on final product strength. In this research, an extrusion temperature of 230 °C was used. This temperature is where the LW-PLA foaming agent starts to act, providing some material savings but leaving mechanical properties unaffected [34].

## 3. Results

### 3.1. Heated Bed Insulation

The heated bed results are shown in Figure 3. The default heated bed power consumption of the Ender-2, Ender-3 and Ender-5 was 28 W, 52 W and 49 W, respectively. When 12 mm thick barafoam was applied, energy consumption was reduced to 26 W, 48 W and 47 W, respectively, a saving of 2.43 W to 4.53 W. The Ender-2 Pro consumed less energy (10%), followed by the Ender-3 (8.6%) and Ender-5 (5%). The reduced power consumption of the Ender-2 compared to the Ender-3 and Ender-5 is due to the lower surface area of the Ender-2 bed. The bed surface area of the Ender-3 and Ender-5 is 484 cm^2^, while the Ender-2 has a bed surface of 272.25 cm^2^. This reduction in surface area decreased heat loss to the environment, thus requiring reduced energy to maintain the constant desired temperature. The observed rate of energy transfer reduction for the three models (5–10%) is significant, particularly as the insulation is a low-cost solution to savings in energy.

### 3.2. Hot-End Insulation

The extruder insulation results are shown in black in Figure 4. The default power consumption of the Ender-2, Ender-3 and Ender-5 was 35.4 W, 34.7 W and 34.4 W, respectively. The consistency between the printers is understandable as all three of these printers share almost identical hot-end configurations. The application of the Creality silicone ‘sock’ reduced consumption to 23.4 W, 23.2 W and 23.8 W, respectively. This is a saving of 11.98 W to 10.52 W. The observed rate of energy transfer reduction for all three models is between 30.63 and 33.8%. This consistency is expected due to the printers having identical hot-end and heater cartridge configurations. Once again this is a significant result in terms of reduced power and the low cost of the printer modification. It is also an increased saving when compared to the heated bed.

### 3.3. Full 3D Printer Enclosure

The most significant absolute power savings were achieved while using the printer enclosures; these results are shown in Figure 5. Over the 8 h time period, the default power consumption of the Ender-2, Ender-3 and Ender-5 was 66.79 W, 88.37 W and 86.29 W, respectively. The application of the enclosure reduced consumption to 56.34 W, 69.60 W and 70.6 W, respectively. This is a saving of 10.45 W to 18.75 W, and the observed rate of energy transfer reduction for all three models is between 15.65 and 18.16%. While using the enclosure, average power consumption fell for all three models, and it was observed that not all models behaved the same. The temperature differential between the air inside the enclosure and outside is time dependent. After five hours, each reached a steady state. The Ender-2 was the lowest, with a steady temperature of 11–12 °C, and the Ender-3 was the highest at 13.5–14.5 °C (Figure 6). The findings show that the increased savings when using an enclosure will be more evident for prints with build times above two hours.

### 3.4. Reduced Support Line Width

Figure 7 shows that the power consumption using a 0.6 mm support line width for the Ender-2, Ender-3 and Ender-5 was 86.88 W, 127.32 W and 118.04 W, respectively. The 0.4 mm width resulted in 86.15 W, 127.03 W and 118.43 W, respectively. The results show that the Ender-2 has the lower energy usage when producing the test part. However, change from a 0.6 mm to 0.4 mm width does not produce a significant saving (less than 1%). The PZEM-017 has a measurement accuracy of ±1% [14]. As such, the energy consumption decreases from the reduced support line width and the lightning infill are marginal as they are close to this threshold. 

As expected, reducing the support width from 0.6 mm to 0.4 mm shows savings in polymer utilisation. For all three machines, the savings amounted to 31.2–32.3%, which can be seen in Figure 8. This is consistent with the CURA slicing software, which predicted a polymer usage of 42 g and 28 g for support widths of 0.6 mm and 0.4 mm, respectively, constant across all three machines. The time saved by using the 0.4 mm line width was below 1 % for all three machines. This is understandable as varying the support line width change does not change the generated path of the hot-end. This is also supported by the CURA slicing software, which predicted a printing time of 2 h and 27 min across all printers regardless of the support line width selected.

### 3.5. Lightning Infill

The power consumption using standard cubic infill for the Ender-2, Ender-3 and Ender-5 was 75.20 W, 92.30 W and 117.7 W, respectively. Using the lightning infill resulted in 112.4 W, 138.00 W and 170.05 W, respectively (Figure 9). This shows that for the lightning infill there was an actual increase in the rate of energy transfer (37.2 W to 52.7 W). This close to 50% increase for all machines is explained by the much faster build time when using the lightning infill. Importantly, when considering the total energy usage independent of build time (J), it can be seen that there is no significant difference between the infill types for all three printers, and thus no significant power savings.

The important saving is on time and material. The lightning infill results in a time saving of 32–34% for the three printers, and the polymer saving is 51% for all three printers when using infill (Figure 10). The choice of this infill method is shown to have little impact upon total energy consumption, but significant savings may be made on time to print and the material consumption for the model. As this modification is most effective on models with a high amount of infill, it may be inferred that models that may benefit from this the most are those with a great volume-to-surface-area ratio.

### 3.6. Optimised Settings and Use of Foaming PLA

Figure 11 shows the power consumption of all three machines with and without optimised settings. The optimised settings include the three insultation adaptations (bed, hot-end and enclosure), and the reduced line width (0.4) and the foaming PLA. The results from producing the test parts show that there is a reduced amount of power for all three machines. The application of the enclosure Ender-2, Ender-3 and Ender-5 reduced consumption to 29.2%, 38.3% and 39.2%, respectively. These are significant savings, and based on the results from the enclosure insulation results, these savings could be even higher with prints requiring a longer build time.

The savings in terms of time and polymer used are also quite significant. For all three machines, a time saving of 21.3–24.5% is observed (Figure 12). There are also differences in the material used with a reduction of 55.7–56.4% when using the optimised settings.

## 4. Discussion

Energy and polymer savings with easily adoptable changes have been demonstrated for the FFF process. To further emphasise the real-life application of these savings, a simple demonstration is made. The Thingiverse website was started in November 2008, and in 2013, Makerbot and Thingiverse were acquired by Stratasys [35]. The site is a large repository of 3D models (>2 million), and for each it shows the download count. By taking the top ten downloads, it possible to work out potential savings in material and power. When considering the polymer savings for these ten products and based on the number of downloads, it is shown that 91.51 t of material could be saved from the adoption of the reduced line support and lightning infill. The power alone for this material saving amounts to 210,000 kWh of energy saved (Table 3). 

This is a conservative estimate of how much energy and material would be saved by the tested modifications, as these are only the top ten models. Additionally, this example has not shown the additional savings that could be made with the insulation optimisation. With the adoption of the demonstrated savings, it is not possible to accurately predict the material and energy savings that could be made for all the different models that are being printed worldwide. However, when one considers that the market for 3D printing filament in 2019 was USD 471.3 million [36] it is clear that the energy and material savings could be significant for a wide range of FFF printers.

## 5. Conclusions

Fused Filament Fabrication can produce a wide range of products. Considered an energy-intensive technique, there are opportunities to improve energy savings to reduce the environmental impact of 3D printing. By using a multi-machine approach, this research has shown a method for identifying thermal management and material usage opportunities to minimise power consumption and provides solutions for a more positive impact on sustainable development. The main conclusions based on the obtained results are:For the insulation research, it was shown that heated bed insulation did not provide exceptional power consumption savings (5–10%). The hot-end insulation was more distinct, with savings of 33.8–30.63%, and significant power savings were also achieved while using the printer enclosures (15.65–18.16%). Importantly, the enclosure insulation findings show that there will be increased savings with build times above two hours. The ease of installation for these three modifications will ensure energy savings over the operating life.It was shown that there are opportunities to reduce polymer usage and electrical energy consumption for 3D printers by the careful selection of print parameters and not simply accepting ‘default’ parameters. By reducing support line width from 0.6 mm to 0.4 mm, material savings amounting to 31.2–32.3% can be seen, and the choice of lightning infill provides a material saving of 51% for all three printers.The combined power consumption and material saving of all three machines with optimised insulation modifications, reduced line width and the foaming PLA were tested on builds of the Utah Teapot test part. With all of the modifications applied, the material consumption was reduced between 55.8% and 56.4%, and an average power consumption reduction of 29 to 38% was observed.

Research in 3D printing can so often be focused on industrial applications. With the uptake of this disruptive technology going beyond the factory environment, it is important not to underestimate the energy and material footprint that these machines have. As shown in the discussion section the adoption of the insulation and material saving changes shown here could provide marked savings in energy and polymer usage. For the ten examples identified it can be estimated that 91.51t of polymers and to 210,000 kWh of energy saved. Anticipating that significantly more printing is being done across the world the savings identified in this research could have a large impact on the environment.

## Figures and Tables

**Figure 1 polymers-15-01874-f001:**
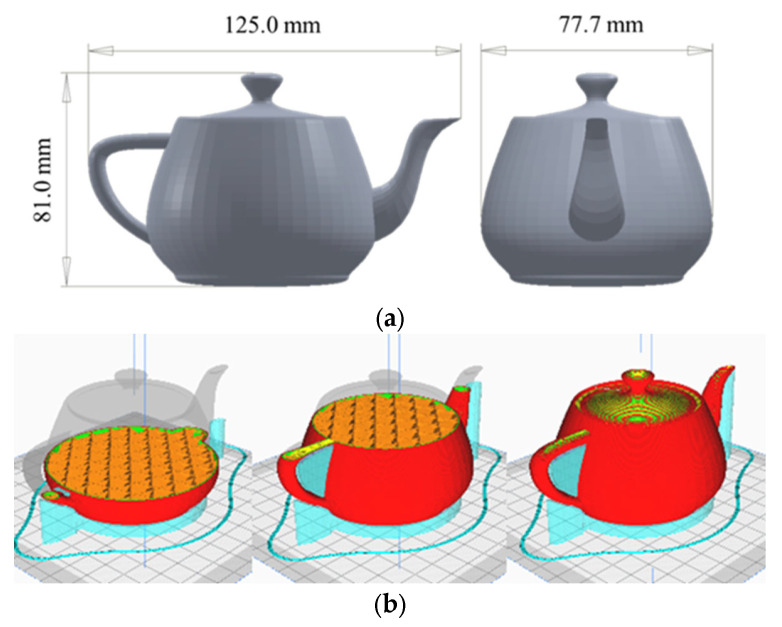
(**a**) Teapot model (**b**) Unmodified g-code (20% Cubic infill, 0.6 mm Support Width, Black colorFabb Economy PLA).

**Figure 2 polymers-15-01874-f002:**
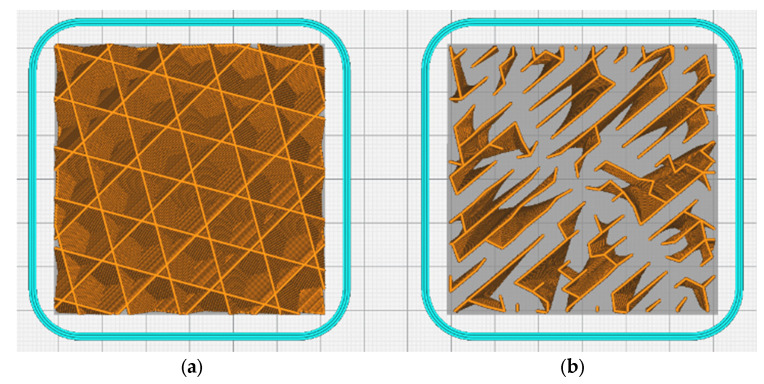
Top View (**a**) 20% Cubic Infill and (**b**) 20% Lightning Infill.

**Figure 3 polymers-15-01874-f003:**
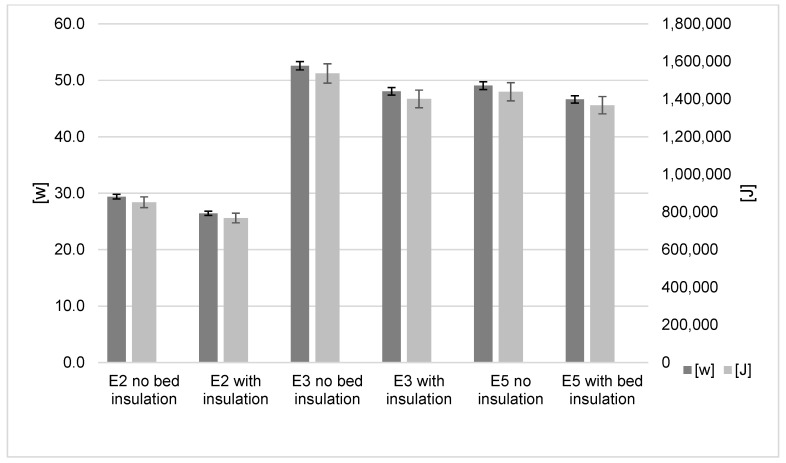
Heated Bed influence on energy results.

**Figure 4 polymers-15-01874-f004:**
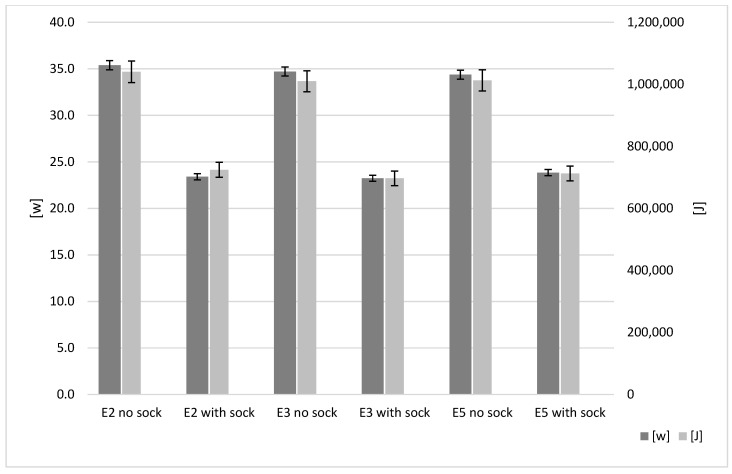
Hot-end influence on energy results.

**Figure 5 polymers-15-01874-f005:**
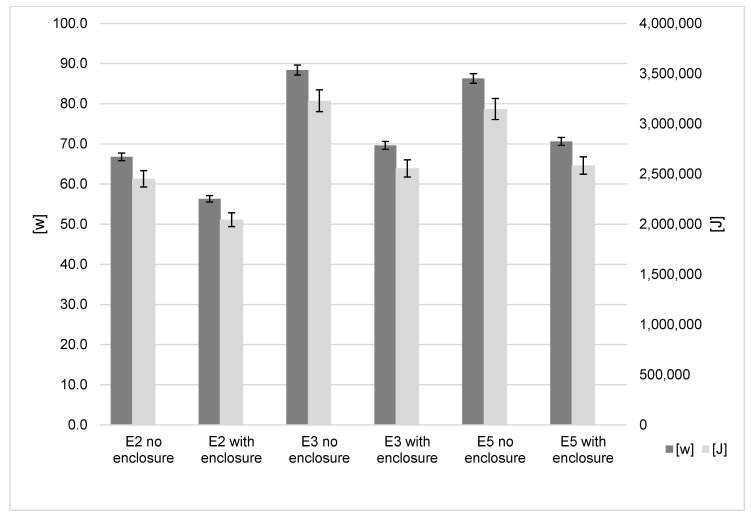
Printer enclosure influence on energy results.

**Figure 6 polymers-15-01874-f006:**
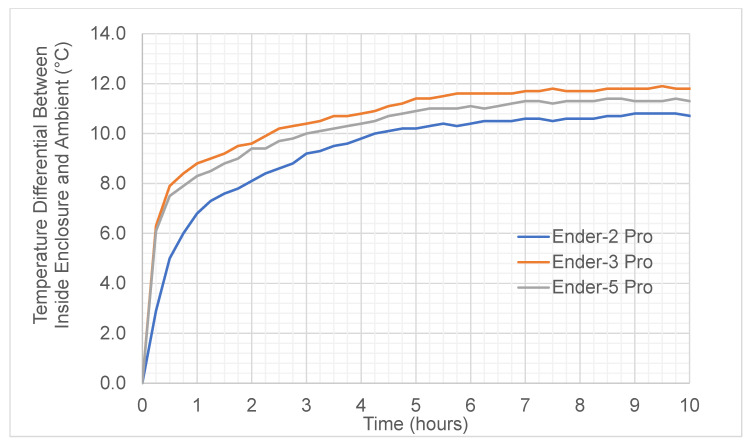
Temperature differential inside and outside the printer enclosure.

**Figure 7 polymers-15-01874-f007:**
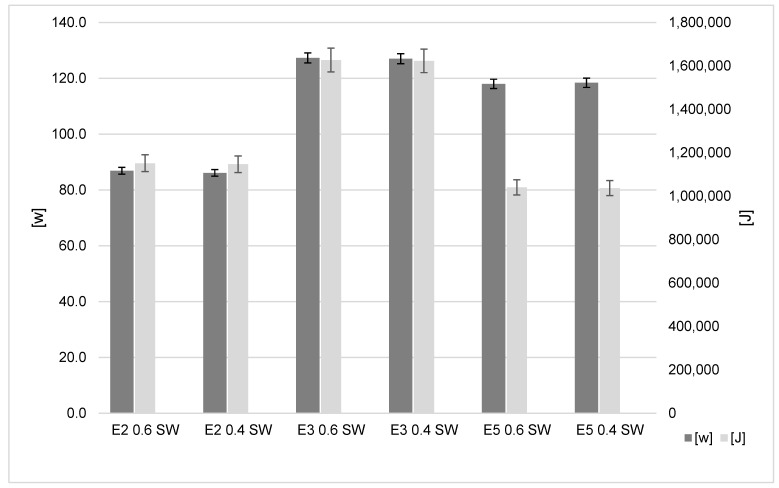
Support line width influence on energy results.

**Figure 8 polymers-15-01874-f008:**
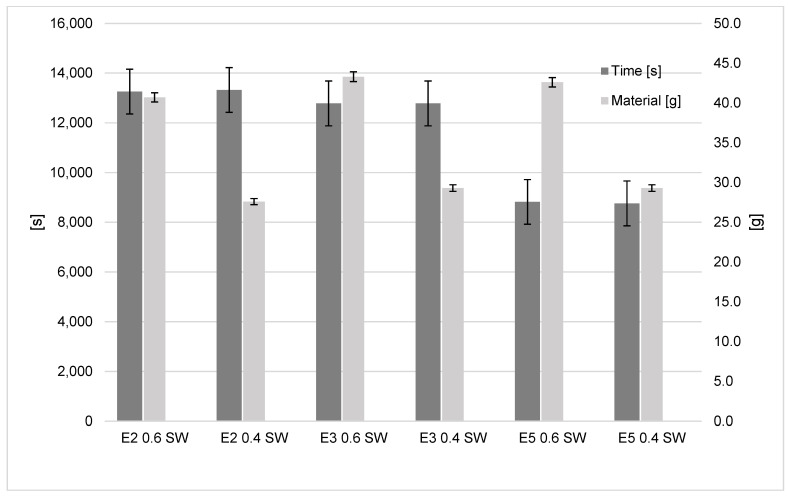
Support line width influence on time and material results.

**Figure 9 polymers-15-01874-f009:**
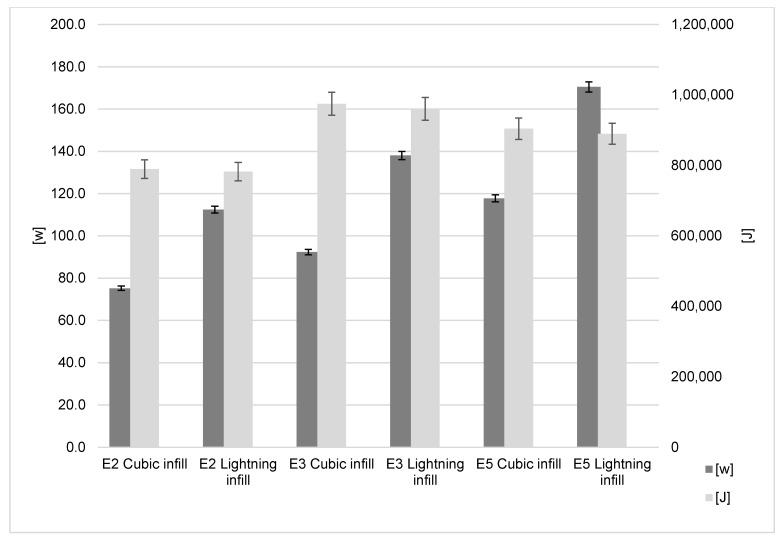
Infill influence on energy results.

**Figure 10 polymers-15-01874-f010:**
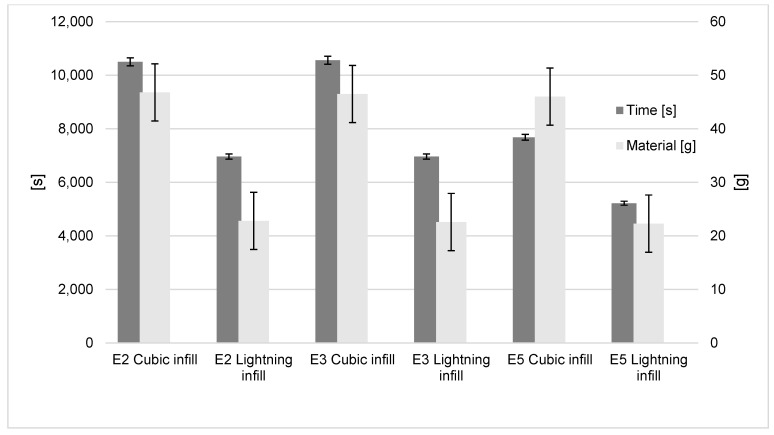
Infill influence on time and material results.

**Figure 11 polymers-15-01874-f011:**
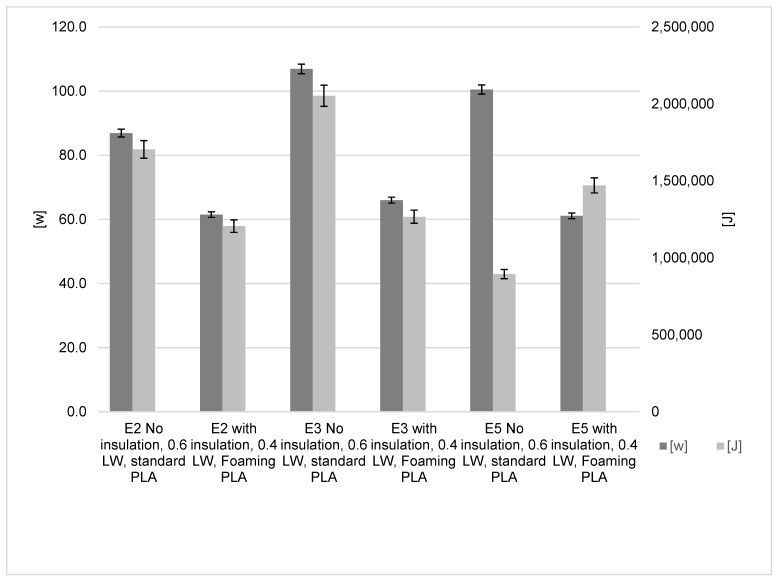
Optimised insulation, line width and foaming PLA influence on energy results.

**Figure 12 polymers-15-01874-f012:**
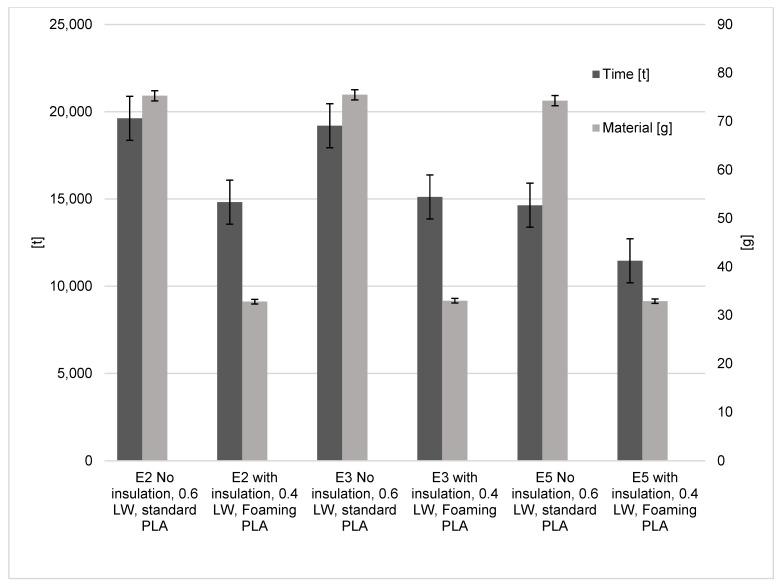
Optimised insulation, line width and foaming PLA influence on material and time results.

**Table 1 polymers-15-01874-t001:** Repeatability analysis.

Experiment	SE Mean (S^2^)	Variance (σ^2^)	COV	Standard Deviation (σ)
1	0.0094	0.00088	0.29	0.0208
2	0.0068	0.00046	0.21	0.0215
3	0.0233	0.006	0.81	0.0237
4	0.0349	0.0097	0.76	0.0297
5	0.014	0.001	0.37	0.0313
6	0.0096	0.00056	0.28	0.0408
7	0.0167	0.0017	0.49	0.0446
8	0.0251	0.0038	0.75	0.0615
9	0.0169	0.002	0.55	0.0773
10	0.0104	0.0004	0.25	0.0987

**Table 2 polymers-15-01874-t002:** Default Ultimaker CURA Printing Settings.

Settings	Default
Layer Height	0.2 mm
Nozzle Diameter	0.6 mm
Line Width	0.6 mm
Support Line Width	0.6 mm
Wall Line Count	2
Top Layers	4
Bottom Layers	4
Infill Density	20%
Infill Pattern	Cubic
Printing Temperature	230 °C
Build Plate Temperature	60 °C
Support Pattern	Zig-Zag
Support Overhang Angle	45°
Connect Support Zig-Zags	On
Support Density	20%
Support Interface	Off
Build Plate Adhesion	Skirt
Skirt Line Count	3

**Table 3 polymers-15-01874-t003:** Materials and power saved.

Product	Material Saving Based on Download Count (Tons)	Power Saving Based on Download Count (kWh)
1	2.58	5920
2	42.56	97,615
3	0.13	301
4	0.01	19
5	0.60	1383
6	0.97	2226
7	38.13	87,443
8	1.54	3531
9	2.43	5571
10	2.56	5863
Total	91.51	209,873

## Data Availability

Not applicable.

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
