# Peer review of "Methods to Reduce Energy and Polymer Consumption for Fused Filament Fabrication 3D Printing"

_polymers, 2023, doi:10.3390/polym15081874_

Round 1

Reviewer 1 Report

Manuscript title: Methods to reduce energy and polymer consumption for fused deposition modelling 3D printing

Manuscript ID: Polymers-2241545

The authors present six techniques for decreasing the environmental impact of the FDM process including thermal insulation methods and material reduction methods. The outcomes identify important thermal management and material consummation opportunities to reduce energy consumption and propose solutions for a more positive impact on sustainable manufacturing of 3D printed parts. The research topics are interesting but the author should clear up the following remarks:

Title:

Within the title, it is mentioned Fused Deposition Modeling. This is the trademark, whereas extrusion-based Additive Manufacturing is the official name with Fused Filament Fabrication (FFF) as a common reference. It is advised to optimize the title.

Abstract:

The names of the process/equipment must be entirely evoked in the text for the first time before being cited as an abbreviation. The Fused Filament Fabrication (FFF) should be remembered at the beginning of the abstract (instead of FDM) and the other parts of the paper.

Keyword:

Because the study is carried out on poly lactic acid (PLA), it’s more relevant to add Poly Lactic Acid (PLA) to the keywords.

1. Introduction

The important point that should be addressed in the introduction paragraph is the differences between the present work and existing studies, either by the same authors or by others.

Page 2: first paragraph, line 58; the author said “In terms of sustainable manufacturing, the FDM printing parameters, namely infill density, layer thickness, feed rate, and shell thickness have influence on the energy usage”.

Other parameters of the FDM process such as build plate adhesion can influence energy usage. The specific energy and the specific time modify by changing the build plate adhesion type and line count. It’s better to revise this paragraph.

2. Experimental

Page 3: second paragraph, line 106; the author said “The three selected 3D printers were Ender-2Pro, Ender-3Pro, Ender-5Pro. These were selected for their range of gantry configurations”.

Technical characteristics such as print volume, printer dimensions and power consumption for the three employed printers (Ender 2, Ender 3, Ender 5) must be mentioned in this section.

Page 3: third paragraph, line 120; the author said “Each was wired as recommended by cutting the heater cartridge cables of the component readings were to be taken from. These leads were connected to the PZEM-017 and a 50A shunt to create a closed circuit”.

Since the operation used to measure the power consumption is quite specific, it is more relevant to illustrate the whole manipulation by adding a photo of the set-up.

Page 3: fourth paragraph, line 137; the author said “ Once all of them had achieved and sustained a bed temperature of 60 °C for at least 60 seconds, a timer of 8 hours was started as the dataloggers were set to start recording results”.

The printing time depends on the size of the piece and the FDM printing conditions, in particular the printing speed, why did you choose a printing time of 8 hours for this manipulation?

Please justify it in the text.

Page 3: fourth paragraph, line 141; the author said “ To assess the effectiveness of this insulation, the methodology described earlier was repeated with the 12 mm thick barafoam in place”.

As described, for each configuration you have carried out an 8-hour test and once. How do you assess the repeatability of the obtained results?

Page 4: first paragraph, line 149; the author said “ Before insulation testing was conducted, each printer was individually turned on and heat block was set to reach 230 °C”.

The default PLA printing temperature offered by Cura software is 200 °C. Why did you choose a temperature of 230 °C for hotend insulation?

Introduce your explanation in the text.

Page 4: second paragraph, line 165; the author said “ the hotends were cleaned using the process described previously in section B”.

Section B should be numbered, revised this sentence.

Page 6: first paragraph, line 259; the author said “ The difference between the two mass readings reflects the total mass of filament consumed. This process was then repeated using g-code where the support line width was reduced to 0.4mm”.

The Cura software can calculate the printing time and the consumed mass for each digital model according to the chosen printing parameters. Can you add the consumed mass to print the Teapot with the support line width of 0.4 mm and 0.6 mm, calculated by the Cura software, in the text?

It will be more relevant to compare these calculated values with those you have measured experimentally.

3. Results and discussion

Page 7: second paragraph, line 265; the author said “The reduced power consumption of the Ender-2 compared to the Ender-3 & Ender-5 is due to a lower surface area of the Ender-2 bed”.

As mentioned above, it is necessary to specify the surface area of the three printers (Ender-2, Ender-3, and Ender-5). Knowing that the surface area of Ender-3 Pro (220 mm * 220 mm * 250 mm) and Ender-5 Pro (220 mm * 220 mm * 300 mm) is the same, why the energy consumption reduction is more important (8.6 %) for the Ender-3 Pro printer?

Page 7: third paragraph, line 275; the author said “The default power consumption of the Ender-2, Ender-3, and Ender-5 was 35.4 W, 34.7 W, and 34.4 W respectively”.

Since the obtained values are very close, it will be better to repeat the manipulation for each configuration and define the standard deviation and the uncertainty of the results.

Figures:

-          Some figures are in color and others are in black and white. It is necessary to harmonize the display mode of all the figures (either in color or black & white).

-          It is necessary to add the error bar on the displayed histograms (figures 3 to 12).

Author Response

Manuscript ID: Polymers-2241545

The authors present six techniques for decreasing the environmental impact of the FDM process including thermal insulation methods and material reduction methods. The outcomes identify important thermal management and material consummation opportunities to reduce energy consumption and propose solutions for a more positive impact on sustainable manufacturing of 3D printed parts. The research topics are interesting but the author should clear up the following remarks:

Comment 1.

Within the title, it is mentioned Fused Deposition Modelling. This is the trademark, whereas extrusion-based Additive Manufacturing is the official name with Fused Filament Fabrication (FFF) as a common reference. It is advised to optimize the title.

Response. Title has been changed to Fused Filament Fabrication throughout the paper.

Comment 2.

Abstract: The names of the process/equipment must be entirely evoked in the text for the first time before being cited as an abbreviation. The Fused Filament Fabrication (FFF) should be remembered at the beginning of the abstract (instead of FDM) and the other parts of the paper.

Response. This has been done.

Comment 3.

Keyword: Because the study is carried out on poly lactic acid (PLA), it’s more relevant to add Poly Lactic Acid (PLA) to the keywords.

Response. Following this advice, we have added Polylactic Acid (PLA) to the paper keywords.

Comment 4.

Introduction. The important point that should be addressed in the introduction paragraph is the differences between the present work and existing studies, either by the same authors or by others.

Response. There are many important papers on FFF and the authors have published several. However, as pointed out in the introduction, Insufficient research conducted on energy use, and accumulative energy impact for FFF. This gap is what drew us to undertake this research and as suspected from many pre-trials the main findings are important.

Comment 5

Page 2: first paragraph, line 58; the author said “In terms of sustainable manufacturing, the FDM printing parameters, namely infill density, layer thickness, feed rate, and shell thickness have influence on the energy usage”. Other parameters of the FDM process such as build plate adhesion can influence energy usage. The specific energy and the specific time modify by changing the build plate adhesion type and line count. It’s better to revise this paragraph.

Response. This is a good point and it has been added to the text.

Comment 6.

  1. Experimental. Page 3: second paragraph, line 106; the author said “The three selected 3D printers were Ender-2Pro, Ender-3Pro, Ender-5Pro. These were selected for their range of gantry configurations”. Technical characteristics such as print volume, printer dimensions and power consumption for the three employed printers (Ender 2, Ender 3, Ender 5) must be mentioned in this section.

Response. The generic power consumption cannot be known, but we have added the following information to the text ‘with a  bed surface area of 484 cm2 for the Ender 3 and Ender 5, and  272.25 cm2 for the Ender 2.’

Comment 7.

Page 3: third paragraph, line 120; the author said “Each was wired as recommended by cutting the heater cartridge cables of the component readings were to be taken from. These leads were connected to the PZEM-017 and a 50A shunt to create a closed circuit”. Since the operation used to measure the power consumption is quite specific, it is more relevant to illustrate the whole manipulation by adding a photo of the set-up

Response. Originally, we included these images but the page number and image number for the journal is at the limit so the editor asked us to remove them.

Comment 8.

Page 3: fourth paragraph, line 137; the author said “Once all of them had achieved and sustained a bed temperature of 60 °C for at least 60 seconds, a timer of 8 hours was started as the dataloggers were set to start recording results”. The printing time depends on the size of the piece and the FDM printing conditions, in particular the printing speed, why did you choose a printing time of 8 hours for this manipulation? Please justify it in the text.

Response. A testing time of 8 hours was chosen to minimize the effects of spikes in the power consumption. These fluctuations are caused by the negative feedback loop the motherboard uses to switch the heater from on to off based on the thermistor temperature. Therefore, an average consumption over a longer period of time is a more representative indicator of energy loss throughout a print.

Comment 9.

As described, for each configuration you have carried out an 8-hour test and once. How do you assess the repeatability of the obtained results?

Response. The results are expected to be repeatable due to the long testing time period and how the ambient conditions were maintained at a constant level throughout that time. We have also added an new section (2.7.4) and table (Table 2) to the paper that includes the repeatability of the results at the pre trial stage.

Comment 10.

Page 4: first paragraph, line 149; the author said “Before insulation testing was conducted, each printer was individually turned on and heat block was set to reach 230 °C”. The default PLA printing temperature offered by Cura software is 200 °C. Why did you choose a temperature of 230 °C for hotend insulation? Introduce your explanation in the text.

Response. To further explain this choice, we have added the following text to the paper.’A hotbed temperature 230 °C to allow for these results to be incorporated and analyzed alongside the foaming polymer testing. As the foaming PLA foams between 215°C and 250°C the setting of 230°C to allow for the foaming filament to have potential impact in its testing while allowing results to be analogous to normal 3D printing settings.’

Comment 11.

Page 4: second paragraph, line 165; the author said “the hotends were cleaned using the process described previously in section B”. Section B should be numbered, revised this sentence.

Response. This has been revised to 2.4.

Comment 12.

Page 6: first paragraph, line 259; the author said “ The difference between the two mass readings reflects the total mass of filament consumed. This process was then repeated using g-code where the support line width was reduced to 0.4mm”. The Cura software can calculate the printing time and the consumed mass for each digital model according to the chosen printing parameters. Can you add the consumed mass to print the Teapot with the support line width of 0.4 mm and 0.6 mm, calculated by the Cura software, in the text?

Response. The consumed mass calculated by CURA is only to the nearest gram; CURA predicts a usage of 42g while using the default support line width of 0.6mm. This prediction is reduced to 28g when the support line width is set to 0.4mm. (Line 368 to 374)

Comment 13.

  1. Results and discussion

Page 7: second paragraph, line 265; the author said “The reduced power consumption of the Ender-2 compared to the Ender-3 & Ender-5 is due to a lower surface area of the Ender-2 bed”. As mentioned above, it is necessary to specify the surface area of the three printers (Ender-2, Ender-3, and Ender-5). Knowing that the surface area of Ender-3 Pro (220 mm * 220 mm * 250 mm) and Ender-5 Pro (220 mm * 220 mm * 300 mm) is the same, why the energy consumption reduction is more important (8.6 %) for the Ender-3 Pro printer?

Response. We have now included surface area of the different heated beds within the text.

Comment 14.

Page 7: third paragraph, line 275; the author said “The default power consumption of the Ender-2, Ender-3, and Ender-5 was 35.4 W, 34.7 W, and 34.4 W respectively”. Since the obtained values are very close, it will be better to repeat the manipulation for each configuration and define the standard deviation and the uncertainty of the results.

Response. The consistency between the printers is understandable as all 3 of these printers share almost identical hotend configurations. Rather than standard deviation & uncertainty being introduced, the similarity of those 3 values functions as evidence of the value reliability. Also see next comment.

Comment 15.

Figures: Some figures are in color and others are in black and white. It is necessary to harmonize the display mode of all the figures (either in color or black & white). It is necessary to add the error bar on the displayed histograms (figures 3 to 12).

Response. The graphs are now harmonised. Regarding the repeatability. Identical 3D printed parts under several conditions was investigated using a Design of experiments approach. This was not added to the paper. In the paper (page 3) we have now added additional text and a table to show the repeatability of the prints. The following text explains this approach. 2.2 Pre-trials. Pre-trials confirmed that the experimental approach of repeated responses for part weight and energy consumption were consistent. However, to validate this a Design of experiments approach was performed. Screening experiments using a ISO 527-2-1A dog bone test part were performed, the control factors included PLA and LW-PLA materials and process factors of , layer height, infill, print speed and extrusion temperature. To establish that the process was in control SE Mean (standard error S2),Variance (σ2) and the Coefficient of Variation (COV) were investigated (Table 1). The results are consistent and the reproducibility shows that the process is optimised for the ten experiments.’

Reviewer 2 Report

The use of the abbreviation is allowed only if this term is defined in the first presentation. FDM should be corrected in the abstract (line 6).

The abstract is very superficially. The abstract should be written more attractively. The novelty of the article should be clearly added to the abstract. Use quantitative results. The outstanding results of the work should be mentioned. In the whole abstract, no quantitative data and results are presented, which is not acceptable at all. At least the percentage of consumption and energy reduction should be provided for each model.

The first paragraph of the introduction is general information and can be omitted or shortened. It is suggested to use stronger literature for the introduction, especially the applications of smart materials in 4D printing and industry 4.0.

Also, the references used is very small. It is suggested to use these sources to strengthen the introduction. (Assessment of controllable shape transformation, potential applications, and tensile shape memory properties of 3D printed PETG --- Development of Pure Poly Vinyl Chloride (PVC) with Excellent 3D Printability and Macro- and Micro-Structural Properties --- Experimental investigation on mechanical characterization of 3D printed PLA produced by fused deposition modeling (FDM) --- A comprehensive experimental investigation on 4D printing of PET-G under bending --- 4D printing of PET-G via FDM including tailormade excess third shape).

In the 3D printing section, all the basic printing parameters should be mentioned. It is suggested to summarize these parameters in a table. Nozzle temperature, bed, printing speed, orientation, raster angle, layer thickness etc. Also, the printed geometry should be specified and their image should be provided.

The research method section is very vague and no specific criteria and standards have been provided for the conducted tests.

Most of the results sections are reports of experimental results that should be improved with deeper discussion and analysis.

How has the reproducibility of the results been checked? Use the scale bar for images. Add error bars to figures.

The conclusion section also needs fundamental corrections, which should be corrected according to the abstract comments.

Author Response

Manuscript ID: Polymers-2241545

Comment 1.

The use of the abbreviation is allowed only if this term is defined in the first presentation. FDM should be corrected in the abstract (line 6).

Response. This has been done throughout the text.

Comment 2.

The abstract is very superficially. The abstract should be written more attractively. The novelty of the article should be clearly added to the abstract. Use quantitative results. The outstanding results of the work should be mentioned. In the whole abstract, no quantitative data and results are presented, which is not acceptable at all. At least the percentage of consumption and energy reduction should be provided for each model.

Response. The point made is a good one. We have rewritten the abstract and included quantitate data that emphasises the importance of the research findings

Comment 3.

The first paragraph of the introduction is general information and can be omitted or shortened. It is suggested to use stronger literature for the introduction, especially the applications of smart materials in 4D printing and industry 4.0.

Also, the references used is very small. It is suggested to use these sources to strengthen the introduction. (Assessment of controllable shape transformation, potential applications, and tensile shape memory properties of 3D printed PETG --- Development of Pure Poly Vinyl Chloride (PVC) with Excellent 3D Printability and Macro- and Micro-Structural Properties --- Experimental investigation on mechanical characterization of 3D printed PLA produced by fused deposition modeling (FDM) --- A comprehensive experimental investigation on 4D printing of PET-G under bending --- 4D printing of PET-G via FDM including tailormade excess third shape).

Response. The introduction paragraph 1 has been edited (169 word to 129). For the point about stronger literature and the suggested research, we agree these streams are of interest to us, particularly the PETG. We are also currently investigating 3d printing of LW-ASA and Varishore TPU, as you know both materials have exiting applications and advance to the process. However, the word count and figure count limitations have meant that more than >1000 words and >8 figures had to be omitted for this particular submission. For this reason, the edit required us to only focus on the main activities presented in the paper.

Comment 4.

In the 3D printing section, all the basic printing parameters should be mentioned. It is suggested to summarize these parameters in a table. Nozzle temperature, bed, printing speed, orientation, raster angle, layer thickness etc. Also, the printed geometry should be specified and their image should be provided.

Response. Originally, we included many more images and figures but the page number and image number for the journal is at the limit so the editor asked us to remove them. We have re added the main image as suggested.

Comment 5.

The research method section is very vague and no specific criteria and standards have been provided for the conducted tests.

Response. This is a good point. In section 2.1 we have summarised the experiments in number form. This will now help the reader understand how the experiments are conducted.

Comment 6.

Most of the results sections are reports of experimental results that should be improved with deeper discussion and analysis.

Response. The results in 3.1 -3.4 all provide a comment on the quantifiable significance of the experiments. Beyond the original brief we also included section 3.5 that highlights the significant savings that could be established when summarizing the observed results. These findings speak for themselves, but to add an emphasis on the importance of the research we included a discussion section. This important additional analysis highlights the wider implications of the established energy and polymer savings. After the edit that has carefully taken into consideration the peer review comments, the paper is improved and we believe that the all of the main research points are well addressed for the reader.

Comment 7.

How has the reproducibility of the results been checked? Use the scale bar for images. Add error bars to figures.

Response. The repeatability of identical 3D printed parts under several conditions was investigated using a Design of experiments approach. This was not added to the paper. In the paper (page 3) we have now added additional text and a table to show the repeatability of the prints. The following text explains this approach.2.2 Pre-trials. Pre-trials confirmed that the experimental approach of repeated responses for part weight and energy consumption were consistent. However, to validate this a Design of experiments approach was performed. Screening experiments using a ISO 527-2-1A dog bone test part were performed, the control factors included PLA and LW-PLA materials and process factors of , layer height, infill, print speed and extrusion temperature. To establish that the process was in control SE Mean (standard error S2),Variance (σ2) and the Coefficient of Variation (COV) were investigated (Table 1). The results are consistent and the reproducibility shows that the process is optimised for the ten experiments.’

Comment 8.

The conclusion section also needs fundamental corrections, which should be corrected according to the abstract comments.

Response. The abstract is changed and the conclusions are modified and the results ar now in alignment. A change has been made by bullet pointing the 3 main findings and this helps the reader separate the energy, material and optimised results. The balance of quantified information and application is clear and the final paragraph clearly emphasises the wider implications of the research.

Reviewer 3 Report

In the present study, the authors have examined six methods of reducing the energy and material consumption of 3D printing. Using different commercial printers each approach was investigated experimentally and the potential savings were quantified. The background of the work is very essential, as it has many applications in industry. The topic is interesting.

The topic of the work falls within the target journal. The work is well organized. The conclusions are credible as there are many experiments. It can be considered for publication after some modifications.

1. In L92, “Each are” is wrong.

2. In L94, the sentence “is shown, this is” is broken.

3. Please modify the sentence in L142.

4. Also in L208.

5. The sentence in L426 is not proper.

6. What is the effect of surface energy on the manufacturing process? The authors may refer to the recent work: Liu et al., A unified analysis of a micro-beam, droplet and CNT ring adhered on a substrate: Calculation of variation with movable boundaries. Acta Mechanica Sinica, 2013, 29(1): 62–72.

7. The work is mainly concerned with experiments. More theoretical analyses and numerical simulations can be done in future.

Author Response

Rev 3

In the present study, the authors have examined six methods of reducing the energy and material consumption of 3D printing. Using different commercial printers each approach was investigated experimentally and the potential savings were quantified. The background of the work is very essential, as it has many applications in industry. The topic is interesting.

The topic of the work falls within the target journal. The work is well organized. The conclusions are credible as there are many experiments. It can be considered for publication after some modifications.

  1. In L92, “Each are” is wrong.

Response. This is now modified

  1. In L94, the sentence “is shown, this is” is broken.

Response. This is now modified

  1. Please modify the sentence in L142.

Response. This is now modified

  1. Also in L208.

Response. This is now modified

  1. The sentence in L426 is not proper.

Response. This is now modified

  1. What is the effect of surface energy on the manufacturing process? The authors may refer to the recent work: Liu et al., A unified analysis of a micro-beam, droplet and CNT ring adhered on a substrate: Calculation of variation with movable boundaries. Acta Mechanica Sinica, 2013, 29(1): 62–72.

Response. This is certainly of interest to us and going forward we will be considering surface energies within the system, particularly the printer bed.

  1. The work is mainly concerned with experiments. More theoretical analyses and numerical simulations can be done in future.

Response. This is a good point and one that we are currently working on. In particular, we are very close to testing a developed mathematical approach for calculating the energy usage based on the model selection.

Round 2

Reviewer 1 Report

The manuscript has been revised considerably. They need to add the error bar on the displayed histograms.  

Author Response

Rev 1

The manuscript has been revised considerably. They need to add the error bar on the displayed histograms.

Response. These are now done.

Reviewer 2 Report

The comments from the pre-review stage have not been answered well, and the article still faces the previous problems.

The innovation of the article should be mentioned in the abstract.

Add the standard deviation to the data presented in the tables and figures. (Pictures 7-12)

In justifying some comments, the limitation of the number of images in the journal (8 images) has been used. In contrast, firstly, the current version of the article has more than eight figures, and secondly, there is no need to add an image to reply to the comment.

Choosing printing parameters is another challenge of the article. How is the layer thickness of 200 microns and nozzle diameter of 600 microns selected? Also, for almost complex geometry.

The results are not analyzed; only a report of the experimental results is presented. Also, the quality of the article and data provided is not suitable for publication in the journal.

Author Response

Rev 2

The innovation of the article should be mentioned in the abstract.

Response. There is no publication that considers the  six methods of reducing the energy and material consumption of 3D printing particularly for three commercial printers. We feel that the abstract accurately captures the research but taking on board your comment have added the following text.  ‘The implementation of a data logging system has identified significant thermal management and material usage opportunities to minimise power consumption, and provides solutions for a more positive impact on sustainable manufacturing of 3D printed parts.’

Add the standard deviation to the data presented in the tables and figures. (Pictures 7-12)

Response. These are now done

Choosing printing parameters is another challenge of the article. How is the layer thickness of 200 microns and nozzle diameter of 600 microns selected? Also, for almost complex geometry.

Response. The Layer heights of 0.2 mm were selected as they’re frequently a recommended ‘normal’ setting for many printers; this is a default as it provides a good balance between time to print and print quality. Nozzle diameters of 0.6 mm can be used for  filaments with particulates which necessitate the use of a wider diameter nozzle to prevent clogging. Using this size nozzle compared to 0.4 mm nozzle widens the scope of the research in terms of materials that can be processed.

The results are not analyzed; only a report of the experimental results is presented. Also, the quality of the article and data provided is not suitable for publication in the journal.

Response. Based on the original comments many improvements have been made. To the wider point the team that conducted the research has a collective 24 years of 3D printing knowledge. One of the team works in industry solely on 3d printing and another has vast experience of outreach to the wider 3D printing community. We do understand the comment regarding analysis. We are currently working on another paper that further examine constituents of this work, the main focus of this paper is not based on analytical research on energy phenomena. Moreover, what we  have shown is a methodological approach for comparative research. Our developed approach to energy and material usage is detailed and provides conclusions that are supported by the results.  The research required a significant contribution of time and comparative analysis, we did this from the offset because we have no doubt that the findings of this paper are important and will raise awareness of the importance of sustainable practice in 3d printing.

Round 3

Reviewer 2 Report

The answers given are not convincing.

Author Response

Reviewer 2 comment. The answers given are not convincing.

Response. We are disappointed to hear this. We have made very reasonable attempts to address your concerns and cannot see where further improvements can be made.